# Target product profiles for novel medicines to prevent and treat preeclampsia: An expert consensus

Annie Ra Mcdougall [1]*, Andrew Tuttle [2], Maya Goldstein [2], Anne Ammerdorffer [3], A. Metin Gülmezoglu [3], Joshua P. Vogel [1,4]

1 Maternal, Child and Adolescent Health Program, Burnet Institute, Melbourne, Australia, 2 Policy Cures Research, Sydney, Australia, 3 Concept Foundation, Geneva, Switzerland, 4 School of Public Health and Preventive Medicine, Monash University, Melbourne, Australia

* annie.mcdougall@burnet.edu.au

**Data Availability Statement:** All data relevant to the study are included in the article or uploaded as supplementary information.

## Abstract

### Background

Preeclampsia and eclampsia are a leading cause of global maternal and newborn mortality. Currently, there are few effective medicines that can prevent or treat preeclampsia. Target Product Profiles (TPPs) are important tools for driving new product development by specifying upfront the characteristics that new products should take. Considering the lack of investment and innovation around new medicines for obstetric conditions, we aimed to develop two new TPPs for medicines to prevent and treat preeclampsia.

### Methods and findings

We used a multi-methods approach comprised of a literature review, stakeholder interviews, online survey, and public consultation. Following an initial literature review, diverse stakeholders (clinical practice, research, academia, international organizations, funders, consumer representatives) were invited for in-depth interviews and an online international survey, as well as public consultation on draft TPPs. The level of stakeholder agreement with TPPs was assessed, and findings from interviews were synthesised to inform the final TPPs. We performed 23 stakeholder interviews and received 46 survey responses. A high level of agreement was observed in survey results, with 89% of TPP variables reaching consensus (75% agree or strongly agree). Points of discussion were raised around the target population for preeclampsia prevention and treatment, as well as the acceptability of cold-chain storage and routes of administration.

### Conclusion

There is consensus within the maternal health research community on the parameters that new medicines for preeclampsia prevention and treatment must achieve to meet real-world health needs. These TPPs provide necessary guidance to spur interest, innovation and investment in the development of new medicines to prevent and treat preeclampsia.

**Funding:** This work was supported by The Bill and Melinda Gates Foundation (INV-023749 to AMG). The funders had no role in study design, data collection and analysis, decision to publish, or preparation of manuscript.

**Competing interests:** The authors have declared that no competing interests exist.

## Introduction

Hypertensive disorders of pregnancy are responsible for approximately 14% of maternal deaths globally, 99% of which occur in low- and middle-income countries (LMICs) [1,2]. Preeclampsia and eclampsia affect 4.6% and 1.4% of pregnant women respectively and account for the majority of maternal deaths and stillbirths due to hypertensive disorders of pregnancy [3]. The underlying aetiology of preeclampsia is incompletely understood; however, it involves abnormal placental development, imbalance in placental angiogenic factors and a pro-inflammatory response leading to uncontrolled maternal hypertension accompanied by either maternal organ failure (usually kidney or liver dysfunction), neurological symptoms and/or fetal growth restriction [4].

There are currently few effective medicines for the prevention and treatment of preeclampsia. In women at risk of preeclampsia (such as women with diabetes, chronic hypertension or a previous history of preeclampsia), low-dose aspirin can reduce the risk of preeclampsia by 10–20% [5]. For women living in regions with low calcium intake, high-dose calcium supplementation can also prevent preeclampsia [6]. In women who develop preeclampsia, anti-hypertensive medications can prevent severe complications, such as stroke or heart failure [7]. Magnesium sulfate is recommended for women with preeclampsia with severe features to prevent or treat eclamptic seizures [8], though some health care professionals lack confidence or knowledge in how to use magnesium sulfate, or consider it a complex or high risk medication to use due to side effects [9]. While birth of the baby and delivery of the placenta can cure preeclampsia, it can still occur postpartum. In light of the global burden and significant morbidity and mortality caused by preeclampsia, there is an urgent need to not only improve implementation of the few available medicines, but also to identify new agents that can prevent or treat it. However, previous research has identified considerable under-investment in in pharmaceutical research specific to maternal health conditions, including preeclampsia [10,11].

The Accelerating Innovation for Mothers (AIM) project was established in 2020 by Concept Foundation, with funding support from the Bill & Melinda Gates Foundation. The AIM project has co-ordinated a number of parallel research activities, including mapping the maternal health medicines development pipeline; identifying the scientific, financial, legal and regulatory barriers to maternal health medicines research; and developing new target product profiles (TPPs) for priority conditions [12].

TPPs are an important resource for funders, researchers, product developers, manufacturers and regulators [13]. They guide product developers on the characteristics required to meet clinical and public health needs. They inform research and development (R&D) strategies, help frame product dossiers, streamline communication with regulatory agencies, and help funders set targets [14]. Therapeutics approved by the US Food and Drug Administration (FDA) that addressed a pre-specified TPP have been linked to more rapid regulatory review [15]. WHO and UNICEF have a number of TPPs publicly available to stimulate development for a range of health products, including vaccines for endemic diseases (such as COVID-19, herpes simplex and malaria), point-of-care diagnostic tests (tuberculosis, yellowfever and HIV), devices for newborn care as well as antibiotics and therapeutics for neglected diseases [16,17]. Despite 295,000 maternal deaths and 1.9 million stillbirths occurring worldwide each year, WHO's Health Product Profile Directory has no TPPs for obstetric conditions. In this study, we used evidence synthesis, interviews, surveys and expert consultations to develop new TPPs to understand the requirements needed to be achieved to drive innovation in medicines for preeclampsia prevention and treatment.

## Methods

A TPP describes the minimum and preferred (or optimal) characteristics of a target product, such as clinical indication, target population, desired efficacy, safety, formulation/presentation and stability and storage [18]. We prepared a study protocol on TPP development, which was informed by methods used in recent TPPs for HIV cures and sexually transmitted infection diagnostic tests, and adopted the five-step process used by Lewin et al. [19–21] This protocol was reviewed and approved by the Alfred Ethics Committee for Human Research (project number 108/21), and formal consent was obtained from all participants prior to their participation.

### Step 1: Initial drafting phase

The AIM project convened a multi-disciplinary expert advisory group, comprising 11 experts from research, obstetrics, patient advocacy, programs implementation, social enterprise, donors, WHO and global health systems with diversity in gender and geographical location. In consultation with this group, developing TPPs for preeclampsia was prioritised amongst five conditions (postpartum haemorrhage, preterm birth, fetal growth restriction and fetal distress), as well as the development of separate TPPs for 1) agents for preventing preeclampsia in women at increased risk; and 2) agents to treat women with preeclampsia.

We then sought to answer the following questions: What would the intended use-case scenario be for prophylactic and therapeutic medicines for preeclampsia? What are the key variables that would need to be considered for TPPs for new medicines to prevent and treat preeclampsia, and what would be the acceptable minimum and preferred targets for each of these variables? Through consultation and literature review, intended use-case scenarios were developed which were revised in subsequent project phases (Box 1). As a guiding principle, given the considerable burden of preeclampsia affecting women in low- and middle-income countries (LMICs), we specified that a primary focus of the TPPs were to drive development of medicines that could be used safely and effectively for women living in limited-resource settings. For example, variables would need to meet the temperature and humidity stability, and route of administration that are practical in low-resource settings. We collated TPP templates

### Box 1. Use case scenarios for TPPs in prevention and treatment of preeclampsia

***Prevention of preeclampsia***:

An affordable drug that can be administered to pregnant women identified as being at increased risk of developing preeclampsia. The drug will prevent the development of preeclampsia, have a good safety profile, can be commenced early in pregnancy (i.e., before 20 weeks' gestation) and can be continued throughout pregnancy, as required.

***Treatment of preeclampsia***:

A therapeutic agent that can be administered by skilled health personnel to pregnant women diagnosed with preeclampsia of any severity, accompanied by monitoring of maternal and fetal well-being in antenatal care settings. The therapeutic agent will delay or prevent maternal disease progression, and ideally improve outcomes for the baby.

and guidance produced by several reputable organizations (FDA, Gates Foundation, WHO, PATH and others) [13,18,22] and developed a TPP template with 21 domains. We conducted literature reviews and developed draft minimum and preferred targets for each domain, along with additional annotations including the rationale and supporting evidence.

## Step 2: International stakeholder survey

In parallel with the stakeholder interviews, we conducted an international online stakeholder survey using the same version of the TPPs. We used the approach of Pelle et al in developing TPPS for new point-of-care diagnostic tests [23]. The survey was conducted using the online survey platform Qualtrics (https://www.qualtrics.com/au/) and was pre-tested on three individuals prior to launch. Respondents were asked to rate their agreement with the minimum and preferred targets for each domain using a Likert scale (1 equals strongly disagree and 5 equals strongly agree). Optional comments were invited for each domain.

The population of interest for the survey was professionals working in the field of maternal and perinatal health. This includes clinicians, researchers, funding agency staff, international public organization staff, programme implementers, policymakers, representatives of consumer advocacy organizations and other relevant maternal health systems stakeholders. Diverse representation from high-, middle- and low-income countries was sought. Survey invitations were sent to approximately 270 individuals using several databases: 1) AIM project database of relevant maternal health R&D experts; 2) a database of all individuals who had participated in WHO maternal and perinatal health guideline development groups in the past 12 years; [24] and 3) members of the WHO Multi-Country Survey on Maternal and Newborn Health research network [25]. To increase participant diversity, we distributed the survey through other clinician-researcher networks and listservs, including the Cochrane Pregnancy and Childbirth network and the Perinatal Society of Australia and New Zealand. The study protocol pre-specified a minimum of 50 responses per domain to evaluate the degree of consensus. We defined agreement as <25% of respondents selecting disagree or strongly disagree for a specific variable.

## Step 3: Stakeholder interview phase

We identified 39 stakeholders from clinical, research, academia, international organizations, funder backgrounds, as well as consumer representatives, with a particular interest on preeclampsia and eclampsia. This sample size was selected to include numerous participants in all stakeholder groups and to ensure gender and geographical diversity. Inclusion criteria for participants was that they had experience in preeclampsia research or clinical practice, maternal medicines development, procurement, or implementation. Stakeholders were identified from a database of individuals who have participated in WHO maternal and perinatal health guideline development, an AIM project database of preeclampsia, maternal medicine and maternal programs implementation experts, and other salient clinical, research, advocacy, and professional networks (such as the International Federation of Gynecology and Obstetrics, the International Confederation of Midwives and the Perinatal Society of Australia and New Zealand). Stakeholders were selected in such a way as to ensure appropriate expertise for the topic of interest, with diversity of gender, geographical and technical expertise.

Stakeholders were initially contacted via email by a member of the research team (AM) and invited to participate in an interview. The goal of these interviews was to seek their input on use-case scenarios and minimum and preferred targets for each TPP domain, across both profiles. Stakeholder interviews were semi-structured through use of a pre-tested interview guide (S1 Appendix). Interviews included discussing the TPP domains sequentially, with particular

emphasis on those relevant to their area of expertise or interest. For example, we focused interviews with obstetricians on target population, clinical efficacy, safety, administration route, among others and programme implementors on stability and shelf life, product presentation, affordability and WHO prequalification. We explicitly sought interviewee's views on applicability across different country contexts. We asked participants to contextualise their feedback on the TPPs in terms of what they considered to be areas of disagreement with current text, and "major" and "minor" issues for revision or clarification in future versions of the TPP.

Interviews were conducted between April–June 2021, over Zoom by an AIM project researcher; they commenced with obtaining informed consent and lasted approximately 60 minutes. All interviews were conducted in English and audio- and video- recorded with permission. Participant feedback on the TPPs was captured through reflexive field notes and cross referenced with the recordings when needed. Participants were invited to submit further written comments after the interview if they wished.

### Step 4: Public consultation

The draft TPPs were made available online for public comment via the Burnet Institute and Concept Foundation websites. The call for public consultation was disseminated via the Burnet Institute Twitter account. The public consultation period lasted approximately 4 weeks (concurrent with the international online survey) and was disseminated via social media.

### Step 5: Synthesis and finalisation

We used a qualitative content analysis approach to analyse the interviews [26], aiming to identify the major and minor issues with the TPP domains, and any challenges to implementing or using the TPPs in practice for product development. We used combined directed and summative content analysis approach, meaning that we used our experience with TPP development and issues related to TPP development to develop an initial coding structure (directed), followed by counting and comparisons of major and minor issues with TPP domains (summative). Issues identified by the interview participants were then classified into key themes and "major" or "minor" concerns. The results of the international online survey were analysed descriptively, with a particular focus on areas where consensus was not reached, defined as variables with >25% of respondents indicating they disagreed or strongly disagreed. Variables where consensus was not achieved were modified based on feedback from expert interviews and survey respondent's comments. The final drafts were shared with the AIM expert advisory group for final comments before finalisation and publication.

## Results

Overall, 23 stakeholders (15 females and 8 males) participated in interviews between April–August 2021. Interviewees were from Africa, Asia/Pacific, Europe, USA and South America (Fig 1A), and included 10 obstetrician/researchers, two neonatologists, two drug development researchers, one WHO staff member, two staff of funding organisations, two midwives, two medicines procurement experts, and two women with lived experience of preeclampsia.

### Survey and public consultation results

The survey was active for 31 days and 46 responses were received. Respondents were across all WHO geographical regions (*AFR* 19.6%, *AMR* 27.5%, *EUR* 15.7%, *SEAR* 11.8%, *WPR* 23.5%, *EMR* 2.0%; Fig 1B). Different professions were represented, including researchers (44.7%), clinicians (doctor, midwife or nurse, 23.4%), epidemiologists or public health specialists (10.6%),

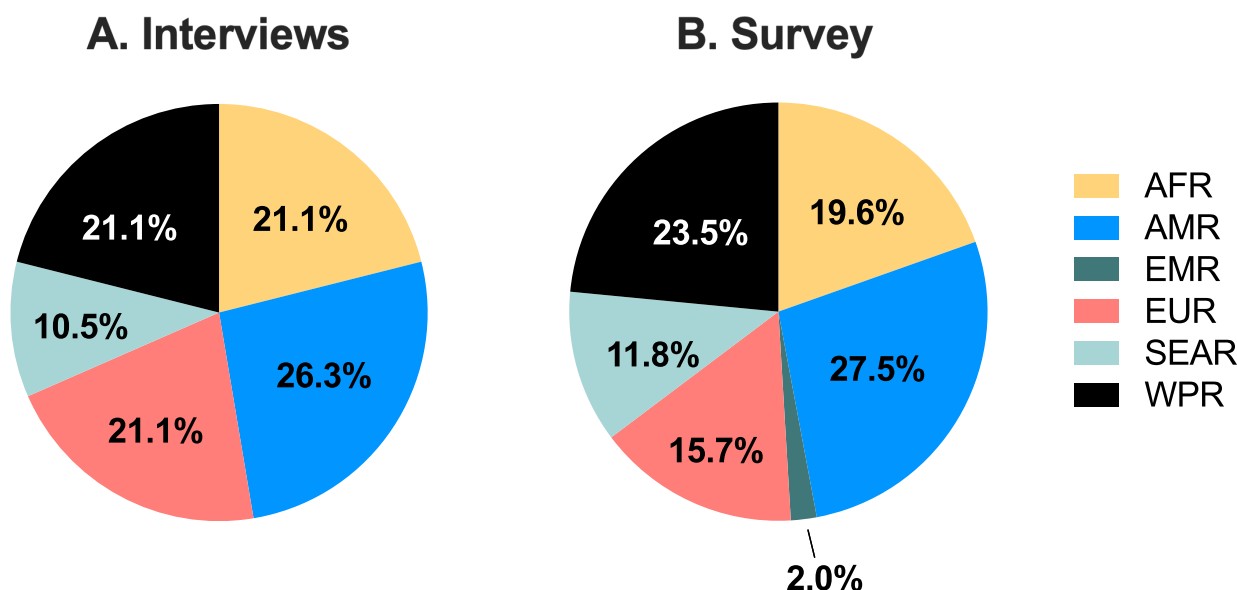

**Fig 1. Stakeholders' distribution by WHO global regions.** The proportion of stakeholders who participated in the (A) interviews (n = 23) and (B) survey (n = 46), in each of the WHO global regions. AFR = African region (yellow), AMR = Region of the Americas (blue), SEAR = South-East Asian region (light green), EUR = European region (red), EMR = Eastern Mediterranean region (dark green), WPR = Western Pacific region (black).

staff or consultants of a national or international NGOs (6.4%), staff of funding agencies (6.4%), employee or consultants of a normative body or civil society organisation (2.1%) or other (including educator, research student and combined clinician/researchers, 6.4%).

Survey results showed high agreement (>75%) across most domains for both TPPs (Fig 2). For preeclampsia prevention, agreement was less than 75% for both the minimum and preferred targets for companion diagnostics; for the minimum targets for administration and stability/shelf-life; and for the preferred target for volume estimates (Fig 2A and 2B). For preeclampsia treatment, agreement was less than 75% for the minimum and preferred targets for population unlikely to be treated; for the minimum targets for stability and shelf-life; and for the preferred target for treatment adherence (Fig 2C and 2D). One comment was received via the public consultation website.

### Findings from stakeholder interviews and survey

Responses to the survey showed a high level of agreement with the target population definition for both prevention (90.2%) and treatment (96.3%). However, a major theme identified during interviews was whether the target population for preeclampsia prevention should be women with identifiable risk factors for preeclampsia versus a population-wide intervention targeting all pregnant women. Some interviewees felt a population-level approach was needed considering the difficulties in accurately predicting preeclampsia [27]. Other interviewees felt that a population-wide intervention required a very high safety profile (complicating or prolonging drug development) and may not be acceptable—most women will not develop preeclampsia, and a population-level prophylaxis approach has significant resource implications. In the TPP for drugs to treat preeclampsia, many interviewees disagreed with the inclusion of women indicated for immediate delivery in the population unlikely to be treated, citing the risk of post-partum preeclampsia. This was consistent with the survey results showing 29.6% of

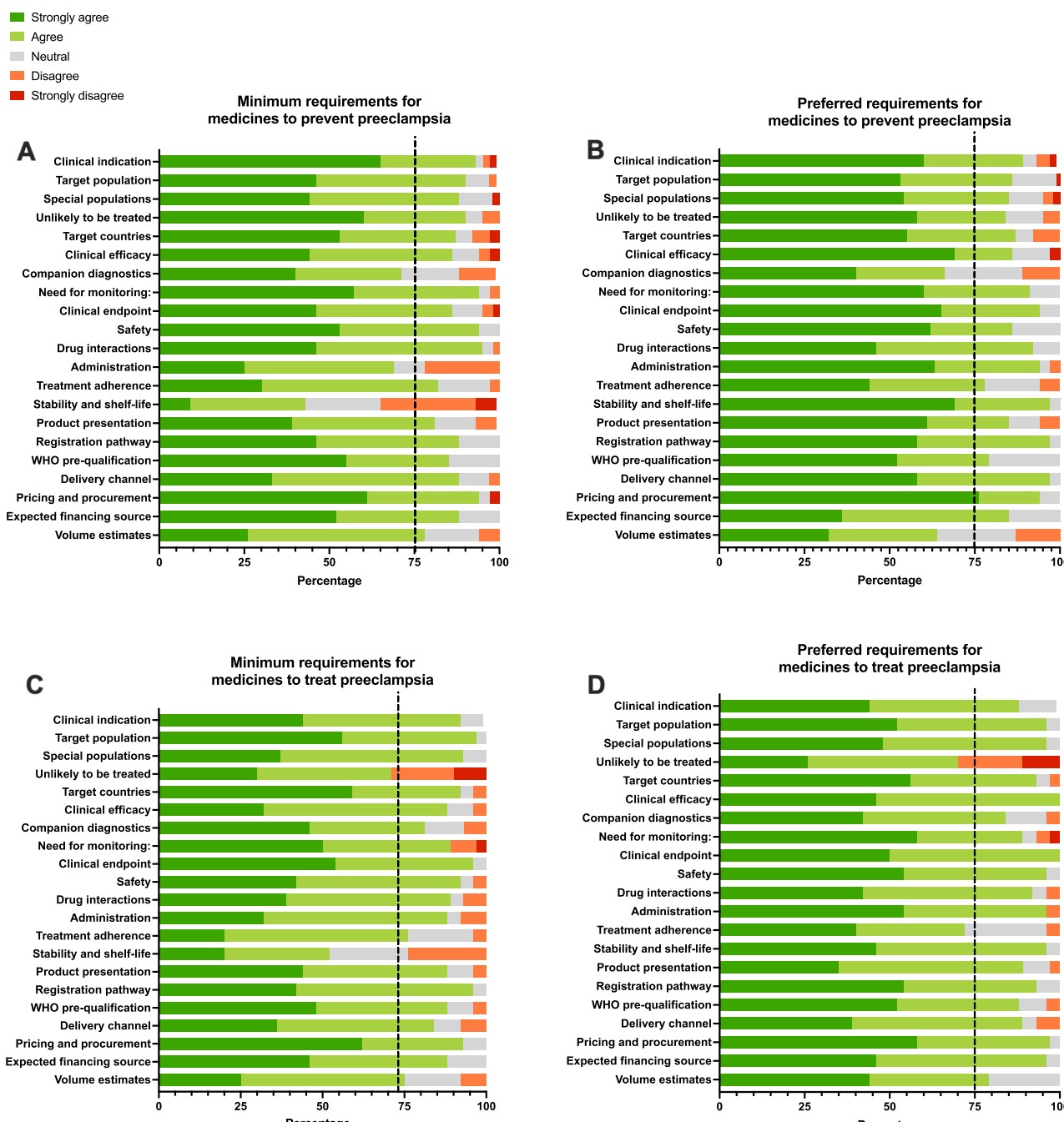

**Fig 2. Survey responses.** Results from international stakeholder survey (n = 46). Percentage of respondents that strongly agreed (dark green), agreed (light green), were neutral (grey), disagreed (orange) or strongly disagreed (red) in response to the minimum and preferred variable in the TPPs for new medicines to prevent (A = minimum criteria, B = preferred criteria) and treat (C = minimum criteria, D = preferred criteria) preeclampsia. Consensus was considered agreement greater than 75% (black dotted line).

respondents either disagreed or strongly disagreed with the minimum and preferred requirements for population unlikely to be treated.

The feedback from the expert interviews on stability and shelf-life, particularly the inclusion of cold chain as a minimum acceptable target also strongly aligned with high disagreement

(24% and 34.4%) in the survey for those domains in both TPPs. Many interviewees, particularly those with experience working in LMICs, highlighted the challenge of maintaining cold chain in these settings. Given the greater burden of preeclampsia in LMICs, ease of transport and storage, as well as stability in hotter or humid conditions is a priority [1]. However, other interviewees maintained that cold chain was an acceptable minimum requirement in order to not limit innovation of novel biological technologies, such as monoclonal antibodies or siRNA therapies [28].

In the TPP for medicines to prevent preeclampsia, many interviewees indicated that they were not comfortable with injectables (i.e., intramuscular or intravenous administration) being included as a minimum requirement for administration, highlighting the greater cost and access barriers if a preventive medicine required injection. Similar to the inclusion of cold chain as acceptable as a minimum in the stability/shelf-life variable, some interviewees felt it was important to include injectable administration as an eligible route of administration within the minimum criteria, so as to not limit the potential innovation around novel biological prevention agents [28]. In all cases, interviewees stated that if injectable was to be included in the minimum requirement, the frequency of injection was vital to its acceptability, with most agreeing that a long-lasting (>1 month) injectable medicine was potentially preferable to once daily oral medication. In addition, multiple interviewees stated that adding administration by non-invasive routes should be considered in both profiles, including options like inhaled, transdermal and vaginal administration. There was some disagreement from other interviewees regarding the inclusion of vaginal administration in specific populations of women. Many interviewees felt that the vaginal route offered benefits over other modes of administration, including localized drug delivery to the uterus, avoiding systemic side-effects, and increased ease of administration compared to injection. In contrast, some interviewees did not support the inclusion of vaginal administration, stating that for some women this would be unacceptable.

In the TPP for medicines to prevent preeclampsia, survey respondents did not reach consensus for the minimum and preferred requirements for companion diagnostics. In contrast, most interviewees agreed with the requirements specified for companion diagnostics. Stakeholders from LMICs specifically highlighted the importance of no special tests being needed to accompany use of the preventive medicine, as this could become a barrier to use in low-resource settings. However, two interviewees proposed that use of newer biomarker tests be included as part of assessing risk factors for preeclampsia. A minor issue identified with both TPPs was that the volume estimates were based on current use of preeclampsia medicines. Multiple interviewees expressed that volume estimates should be based on the incidence of preeclampsia and its risk factors rather than current use of medicines, given the inability to accurately assess current usage of medicines for preeclampsia.

## Finalisation of TPPs

Following synthesis of findings from stakeholder interviews and surveys, the study group agreed that the target population for prevention of preeclampsia would remain as "women at increased risk of preeclampsia" (Table 1). In the TPP for medicines to treat preeclampsia, "women indicated for immediate delivery" was removed from the women unlikely to be treated (Table 2). Text acknowledging post-partum preeclampsia was also added throughout the TPPs. In both TPPs, the inclusion of cold-chain storage was removed from the minimum and preferred requirements and replaced with acceptability of cold-chain for biologicals only. Additional non-invasive administration routes—including transdermal, inhaled, and vaginal routes—were added to the minimum and preferred formulation, dosage and administration.

Table 1. Target product profile for medicines to prevent preeclampsia.

| | Minimum<br>*The minimal target should be considered as a potential go/no go decision point.* | Preferred<br>*The preferred (or optimistic) target should reflect what is needed to achieve broader, deeper, quicker global health impact.* | Annotations<br>*For all parameters, include here the source data used and rationale for why this feature is important.* |
|---|---|---|---|
| **Indication** | Prophylactic treatment of pregnant women at increased risk of developing preeclampsia. | Same as minimum | The medicine is intended to prevent preeclampsia in pregnant women at increased risk, to improved maternal and fetal/neonatal mortality and morbidity outcomes. |
| **Target population** | Pregnant women with identified risk factors for preeclampsia. | Same as minimum | There is currently a lack of consensus on the criteria for identifying women at risk of preeclampsia. WHO recommendations [6] define the risk factors as:<br>*moderate risk*: any two of the following risk factors: primiparity, family history of preeclampsia, age greater than 40 years, or multiple pregnancy.<br>*high risk*: one or more of the following risk factors: diabetes, obesity, chronic or gestational hypertension, renal disease, autoimmune disease, positive uterine artery Doppler, previous history of preeclampsia, or previous fetal/neonatal death associated with preeclampsia.<br>The recommendations note that this not an exhaustive list of risk factors and can be adapted based on the local epidemiology of preeclampsia. |
| **Special populations** | Safe and effective in women with common co-morbidities (e.g., chronic hypertension, type I or II diabetes, obesity, chronic kidney disease or autoimmune disease) and in pregnant adolescents (<18 years old). | Safe and effective in all pregnant women. | The population of women at increased risk of preeclampsia are also more likely to have other co-morbidities, including chronic hypertension, type I or II diabetes, chronic kidney disease or autoimmune disease. |
| **Population unlikely to be treated** | Women with a medical contraindication to the preventive agent.<br>Women currently diagnosed with preeclampsia or eclampsia. | Same as minimum. | - |
| **Target countries** | All high-, middle- and low-income countries | Same as minimum | The incidence of preeclampsia and eclampsia is estimated at 4.6% and 1.4% of pregnant women, respectively [3].<br>Approximately 16% of pregnant women in the UK are at an increased (moderate–high) risk of preeclampsia [29]. |
| **Efficacy** | Clinically significant reduction in preeclampsia incidence, or delayed onset of preeclampsia in women at increased risk. | Clinically significant reduction in preeclampsia incidence, or delayed onset of preeclampsia in women at increased risk.<br>Clinically significant reduction in serious adverse maternal or fetal/neonatal outcomes associated with preeclampsia | WHO recommends that women at moderate or high risk of preeclampsia should be treated with daily low-dose aspirin as a preventive therapy. Based on evidence from 60 studies, aspirin probably reduces the risk of preeclampsia by 18% (RR 0.82, 95% CI 0.77–0.88) [5]. |

*(Continued)*

**Table 1.** (Continued)

| | Minimum<br>*The minimal target should be considered as a potential go/no go decision point.* | Preferred<br>*The preferred (or optimistic) target should reflect what is needed to achieve broader, deeper, quicker global health impact.* | Annotations<br>*For all parameters, include here the source data used and rationale for why this feature is important.* |
|---|---|---|---|
| **Is companion diagnostic needed for use?** | No. Identifying women at risk of preeclampsia requires a thorough history and clinical examination.<br>Some conditions that increase risk of preeclampsia require use of special tests.<br>*moderate risk*: any two of the following risk factors: primiparity, family history of preeclampsia, age greater than 40 years, or multiple pregnancy.<br>*high risk*: one or more of the following risk factors: diabetes, obesity, chronic or gestational hypertension, renal disease, autoimmune disease, positive uterine artery Doppler, previous history of preeclampsia, or previous fetal/neonatal death associated with preeclampsia. | Same as minimum. | A number of risk factors have been identified as increasing risk of preeclampsia many of which are identified based on history and examination, though some (such as gestational diabetes or positive uterine artery Doppler, angiogenic factors) require special tests. |
| **Need for clinical monitoring?** | Regular clinical assessments as part of standard care for women at risk of preeclampsia, including monitoring for fetal health and well-being.<br>Minimal additional monitoring required for expected drug side-effects. | Regular clinical assessments as part of standard care for women at risk of preeclampsia, including monitoring for fetal health and well-being.<br>No additional monitoring required for expected drug side-effects. | Women at risk of preeclampsia should be regularly assessed in antenatal care settings to identify signs or symptoms of preeclampsia. |
| **Clinical endpoint for licensure** | Reduced incidence of preeclampsia amongst pregnant women at increased risk | Reduced incidence of preeclampsia<br>Reduced incidence of adverse maternal and fetal/neonatal outcomes associated with preeclampsia. | Clinical endpoints have been selected based on primary outcomes in Cochrane reviews of current preventative treatments for preeclampsia, and priority outcomes used in WHO guidelines on preventing preeclampsia [5,6]. |
| **Safety** | No significant clinical adverse effects for mother and baby.<br>Not contraindicated in pregnant and lactating women.<br>Absence of embryo-fetal toxicity or teratogenicity. | No clinical adverse effects for mother and baby.<br>No drug-related serious adverse events for mother or baby.<br>Not contraindicated in pregnant and lactating women.<br>Absence of embryo-fetal toxicity or teratogenicity.<br>Evidence shows no long-term adverse effects for mothers or babies. | |
| **Drug interactions** | No significant drug-drug interactions with common antenatal treatments (medicines or supplements), medicines used in women with preeclampsia (such as anti-hypertensives, antibiotics, magnesium sulfate, tocolytics or corticosteroids), or drugs used for common co-morbidities (including chronic hypertension, type I or II diabetes, obesity, chronic kidney disease or autoimmune disease). | No drug-drug interactions with common antenatal treatments (medicines or supplements), medicines used in women with preeclampsia (such as anti-hypertensives, antibiotics, magnesium sulfate, tocolytics or corticosteroids) or drugs used for common co-morbidities (including chronic hypertension, type I or II diabetes, obesity, chronic kidney disease or autoimmune disease). | Preventive agent will be used alongside usual antenatal care for women at increased risk of preeclampsia. Hence, the treatment must have minimal to no adverse interactions with drugs commonly used in pregnant women and women with preeclampsia. |
| **Formulation dosage & administration** | Non-invasive (including oral, inhaled, vaginal or transdermal) or injectable (IM or SC).<br>Treatment can be commenced early in pregnancy (e.g.: prior to 20 weeks' gestation) and can be continued throughout pregnancy, and into the postpartum period, as required.<br>Regimen (dose and duration) dependent on clinical response to preventive agent. | Oral<br>Treatment can be commenced early in pregnancy (e.g.: prior to 20 weeks' gestation) and can be continued throughout pregnancy and into the postpartum period, as required.<br>Regimen (dose and duration) dependent on clinical response to preventive agent. | Current therapies are orally self-administered. Current novel technologies and therapies in development for preeclampsia prevention include non-systemic, targeted, injectables [28]. Oral administration is preferred, and will promote acceptability, self-administration and adherence in line with current therapies. Oral administration would likely be more feasible and acceptable for low-resource settings. |

(*Continued*)

**Table 1.** (Continued)

| | Minimum<br>*The minimal target should be considered as a potential go/no go decision point.* | Preferred<br>*The preferred (or optimistic) target should reflect what is needed to achieve broader, deeper, quicker global health impact.* | Annotations<br>*For all parameters, include here the source data used and rationale for why this feature is important.* |
|---|---|---|---|
| **Treatment adherence** | Frequency of discontinuation during therapy <30% | Frequency of discontinuation during therapy <20% | Large multi-center trials of aspirin and calcium supplements during pregnancy have reported that high intake adherence rates (>80–90%) are required for improved health outcomes. Discontinuation rates are reported as <20% [30,31]. Treatment adherence rates do not take into consideration access to healthcare services or supplies. |
| **Stability / Shelf life** | Stable at 30˚C.<br>Easy to transport and store.<br>2-year shelf life in climatic zone IVb (simulated with 30˚C and 75% relative humidity).<br>*Biologicals*: cold-chain (2-8˚C) acceptable. | Stable at 30˚C.<br>Easy to transport and store.<br>3 to 5-year shelf life in climatic zone IVb (simulated with 30˚C and 75% relative humidity, plus 6 months at 40˚C and 75% relative humidity).<br>*Biologicals*: cold-chain (2-8˚C) acceptable. | Given the greater burden of preeclampsia in LMICs, ease of transport and storage, as well as stability in hotter or humid conditions is a priority [1]. |
| **Product presentation** | Easy to open and administer.<br>Packaging must aim to protect and preserve the quality of the product and prevent damage to the drugs during transport and storage.<br>*Injectable*: packaging must maintain sterility. | Compact, lightweight, easy to open and administer, sustainable packaging.<br>Packaging must aim to protect and preserve the quality of the product and prevent damage to the drugs during transport and storage.<br>*Injectable*: packaging must maintain sterility.<br>Environmental impact of the packaging should be minimized. | An easy to open and administer presentation will aid in the implementation of the preventive agent, as there will be minimal additional training requirements for healthcare workers or women to self-administer.<br>Packaging and design must comply with regulatory guidance from a stringent regulatory authority or WHO standards. |
| **Target product registration pathway (s)** | Approval by at least 1 stringent regulatory authority (e.g., US Food and Drug Administration, European Medicines Agency)<br>Approval from relevant national regulatory authorities will also be required | Approval by at least 1 stringent regulatory authority (e.g., US Food and Drug Administration, European Medicines Agency)<br>Approval from relevant national regulatory authorities will also be required<br>WHO pre-qualification approval obtained | Use of a preventive agent in a given LMIC will require approval from their national regulatory authority.<br>Product registration pathways are likely to differ for repurposed compared to novel drug treatments.<br>Engaging with regulatory authorities early to discuss potential regulatory pathways and streamline the approval process is advised. |
| **WHO prequalification** | WHO listed authority application pathways within 12 months of Essential Medicines List (EML) inclusion. | WHO prequalification submission to be made within 12 months of Essential Medicines List (EML) inclusion. | WHO PQ eligibility follows guideline and EML inclusion. |
| **Primary target delivery channel** | *All*: Antenatal, childbirth and postpartum care settings (including community healthcare settings) where women at risk of preeclampsia receive care.<br>*Non-invasive*: Staff available to provide and advise women on using medicine correctly<br>*Injectable*: Staff, supplies and equipment available and authorised to administer medicine | *All*: Antenatal, childbirth and postpartum care settings (including community healthcare setting) where women at risk of preeclampsia receive care.<br>*Oral*: Staff available to provide and advise women on using medicine correctly | It is anticipated that the preventive agent will be used in antenatal care settings, particularly those where higher-risk women receive care. |
| **Target affordable pricing / procurement** | Preventive agent is affordable in the public sector in LMICs | Preventive agent is affordable in the public sector in LMICs<br>Unit cost of treatment is similar to other preventative therapies for women at increased risk of preeclampsia | Given the burden of preeclampsia in LMICs, affordability of any novel treatments is a high priority.<br>Current preventive agents for women with preeclampsia (aspirin; calcium supplements) are generally widely available and affordable. |

(*Continued*)

**Table 1.** (Continued)

| | Minimum<br>*The minimal target should be considered as a potential go/no go decision point.* | Preferred<br>*The preferred (or optimistic) target should reflect what is needed to achieve broader, deeper, quicker global health impact.* | Annotations<br>*For all parameters, include here the source data used and rationale for why this feature is important.* |
|---|---|---|---|
| **Expected financing source** | Procurement in LMICs financed by national governments, international agencies (including UN organizations), and /or international donors, or private sector | Procurement financed by national governments or private sector | Procurement of medicines for use in pregnancy in LMICs varies between countries, but it may include governments as well as support from international organizations, agencies or funders.<br>For a new treatment, initial support from international organizations maybe required. Procurement of effective treatments would ideally be prioritized by national governments. |
| **Volume estimates** | Volumes compatible with incidence of preeclampsia | Same as minimum | The estimated global incidence of preeclampsia is approximately 5%, equating to ~7 million women worldwide each year (though this may be an underestimate) [3].<br>Limited data exists on the proportion of women who are at increased risk of preeclampsia, however, observational data from the UK report 16.1% of pregnant women have identified risk factors for preeclampsia [29]. There are currently no reliable global estimates on the coverage of current preventative therapies for preeclampsia, though they are widely used. |

## Discussion

Although TPPs have been widely used to stimulate innovation for high-burden health conditions (such as infectious diseases), these are the first TPPs to be developed for new obstetric medicines. Through extensive consultation with a wide variety of experts and stakeholders, we have defined the key characteristics that new medicines to prevent and treat preeclampsia should take, with an emphasis on meeting the needs of women living in LMICs. These TPPs will guide researchers, product developers and industry partners to achieve improvements in current therapies available to women at risk of and with preeclampsia.

During stakeholder interviews we identified disagreement on whether a population-wide (i.e. all pregnant women) or a targeted (i.e. a subset of pregnant women, such as women at increased risk) intervention is likely to be a more effective strategy for preventing preeclampsia. Predicting who will develop preeclampsia accurately is challenging—screening for preeclampsia using clinical risk factors, as proposed by the National Institute for Health and Care Excellence (NICE) and the American College of Obstetricians and Gynecologists (ACOG) have poor predictive value. The NICE guidelines have a detection rate of 41% and 34% for preterm and term preeclampsia respectively, whereas the ACOG guidelines have a detection rate of only 5% of preterm and 2% term preeclampsia [39,40]. This means 59–98% of women who develop preeclampsia are not identified with these screening algorithms. Some improvements in preeclampsia prediction algorithms have been made, particularly with the inclusion of biomarkers [41] however as these are not yet widely used—particularly in LMICs—they were not included as a minimum requirement in these profiles. A further consideration is that most of the promising candidates currently being investigated for prevention of preeclampsia would not be feasible as a population-wide intervention [28]. Preeclampsia diagnostics may themselves be a strong candidate for new TPPs in the future [42].

Table 2. Target product profile for medicines to treat preeclampsia.

| | Minimum<br>*The minimal target should be considered as a potential go/no go decision point.* | Preferred<br>*The preferred (or optimistic) target should reflect what is needed to achieve broader, deeper, quicker global health impact.* | Annotations<br>*For all parameters, include here the source data used and rationale for why this feature is important.* |
|---|---|---|---|
| **Indication** | Treatment of women with suspected or confirmed preeclampsia, regardless of severity. | Same as minimum | A therapeutic target is intended to treat preeclampsia in pregnant or postpartum women, and improve maternal, fetal and/or neonatal mortality and morbidity outcomes. Typically, more severe disease is associated with worse outcomes for mother and baby. Treatment initiated early in disease progression (e.g., in women with mild disease) could potentially have greater benefits. |
| **Target population** | Pregnant and postpartum women with suspected or confirmed preeclampsia, regardless of severity | Same as minimum | ICD-11 characterises preeclampsia as the new onset of hypertension (systolic blood pressure ≥140 mmHg and/or diastolic blood pressure ≥90mmHg) and proteinuria OR significant end-organ dysfunction after 20 weeks of gestation.<br>As the resources for diagnosing preeclampsia may not always be available (particularly in low-resource settings), an agent that is effective in women with suspected preeclampsia would be more practical to implement across LMICs. |
| **Special populations** | Safe and effective in women who are candidates for immediate delivery (for example, those with severe symptoms of preeclampsia or eclampsia, fetus showing signs of distress or severe IUGR), treated to prevent postpartum preeclampsia.<br>Safe and effective in women with common co-morbidities (e.g., chronic hypertension, type I or II diabetes, obesity, chronic kidney disease or autoimmune disease), and in pregnant adolescents (<18 years old). | Safe and effective in women who are candidates for immediate delivery (for example, those with severe symptoms of preeclampsia or eclampsia, fetus showing signs of distress or severe IUGR), treated to prevent post-partum preeclampsia.<br>Safe and effective in all pregnant or postpartum women with any form of preeclampsia, including those diagnosed with HELLP syndrome or preeclampsia superimposed upon chronic hypertension.<br>Safe and effective in women with common co-morbidities (e.g., chronic hypertension, type I or II diabetes, obesity, chronic kidney disease or autoimmune disease), and in pregnant adolescents (<18 years old). | The target product profile for novel preeclampsia treatment is already targeting to a "special population"–pregnant and postpartum women. The optimal requirements would deliver a safe and effective treatment for preeclampsia in all women, including those with HELLP syndrome or preeclampsia superimposed upon chronic hypertension or other medical conditions allowing for delivery of the intervention in settings where the accurate differentiation between preeclampsia subtype was not efficient. |
| **Population unlikely to be treated** | Women with a medical contraindication to the therapeutic agent. | Same as minimum | |
| **Target countries** | All high-, middle- and low-income countries | Same as minimum | The incidence of preeclampsia and eclampsia is estimated at 4.6% and 1.4% of pregnant women, respectively [3]. |
| **Efficacy** | Clinically significant difference in extending pregnancy duration to increase fetal maturity in women with preterm preeclampsia.<br>OR<br>Clinically significant reduction in serious adverse maternal antenatal or postpartum outcomes associated with preeclampsia disease progression (such as mortality, severe-preeclampsia, eclampsia, stroke, etc.);<br>OR<br>Clinically significant reduction in adverse fetal/neonatal outcomes associated with preeclampsia, (such as stillbirth, IUGR, preterm birth, neonatal mortality, admission to the NICU or other preeclampsia-related neonatal complications). | Clinically significant difference in extending pregnancy duration to increase fetal maturity in women with preterm preeclampsia.<br>AND<br>Clinically significant reduction in serious adverse maternal antenatal or postpartum outcomes associated with preeclampsia disease progression (such as mortality, severe-preeclampsia, eclampsia, stroke, etc.);<br>AND<br>Clinically significant reduction in adverse fetal/neonatal outcomes associated with preeclampsia, (such as stillbirth, IUGR, preterm birth, neonatal mortality, admission to the NICU or other preeclampsia-related neonatal complications). | Efficacy outcomes have been selected based on priority outcomes in the WHO guidelines for treating women with preeclampsia, and the core outcome set for preeclampsia [6,32]. |

*(Continued)*

**Table 2.** (Continued)

| | Minimum<br>*The minimal target should be considered as a potential go/no go decision point.* | Preferred<br>*The preferred (or optimistic) target should reflect what is needed to achieve broader, deeper, quicker global health impact.* | Annotations<br>*For all parameters, include here the source data used and rationale for why this feature is important.* |
|---|---|---|---|
| **Is companion diagnostic needed for use?** | The International Classification of Diseases (ICD-11) describes preeclampsia as characterised by systolic blood pressure greater than 140mmHg or diastolic blood pressure greater than 90mmHg on two occasions, 4 hours or more apart in the presence of either proteinuria or other new onset maternal organ dysfunction, neurological conditions or fetal growth restriction.[38]<br>Proteinuria testing or special tests for organ dysfunction may be required for diagnosis. | Same as minimum | Special tests may be required for preeclampsia to be diagnosed.<br>Proteinuria is diagnosed through urinalysis for protein in urine. Additional diagnostic tests include laboratory evaluation of platelet count, serum creatine and liver chemistries [33]. Other special tests (such as placental angiogenic factor-based testing) may be used for preeclampsia diagnosis in some settings. However, these are not widely available in LMICs and should not be regarded as a minimum requirement. |
| **Need for clinical monitoring** | Continued monitoring of maternal, fetal and neonatal health and well-being. For women treated in the postpartum period only, additional monitoring of newborns (beyond routine practice) is not required.<br>Minimal additional monitoring required for expected drug side-effects. | Continued monitoring of maternal, fetal and neonatal health and well-being. For women treated in the postpartum period only, additional monitoring of newborn (beyond routine practice) would not be required.<br>No additional monitoring required for expected drug side-effects. | Expectant management of women with preeclampsia includes regular monitoring of maternal blood pressure, as well as platelet count, serum creatinine and liver chemistries. Fetal growth and well-being also needs to be regularly assessed [6]. |
| **Clinical Endpoint for Licensure** | Clinically important difference in extending pregnancy duration to increase fetal maturity.<br>Reduced maternal clinical endpoints: death or major maternal morbidity (eclampsia, recurrent seizures, stroke, Pulmonary oedema, emergency caesarean, placental abruption etc.)<br>Reduced fetal/neonatal endpoints: stillbirth, neonatal death or major neonatal morbidity (IUGR, preterm birth, low birthweight, NICU admission, respiratory distress syndrome Intraventricular haemorrhage, etc.) | Same as minimum | Clinical endpoints have been selected based on priority outcomes in the WHO guidelines for treating women with preeclampsia, and the preeclampsia core outcome set [6,32]. |
| **Safety** | Clinical safety (adverse or serious adverse effects for mother and baby) comparable to current therapies.<br>Not contraindicated in pregnant and lactating women.<br>Absence of fetal toxicity. | Fewer adverse effects than current therapies.<br>No drug-related serious adverse events for mother or baby.<br>Not contraindicated in pregnant and lactating women.<br>Absence of fetal toxicity.<br>Evidence shows no long-term adverse effects for mothers or babies. | Current treatments for specific manifestations of preeclampsia include antihypertensive drugs (e.g., methyldopa or labetalol) and magnesium sulfate. Drug options recommended by WHO for managing hypertensive disorders or pregnancy largely have acceptable safety profiles, though some lack evidence for fetal safety outcomes [34,35]. Side effects of different anti-hypertensive drugs in pregnancy vary. For example, beta-blockers can cause oedema, postural hypotension, bradycardia, cold extremities, rashes, sweating, tachycardia, nausea, dyspepsia, vomiting and difficulty in micturition [36]. Side effects of magnesium sulfate include flushing, nausea and/or vomiting, slurred speech, muscle weakness, hypotension, dizziness, drowsiness or confusion, and headache [8]. |

*(Continued)*

**Table 2.** (Continued)

| | Minimum<br>*The minimal target should be considered as a potential go/no go decision point.* | Preferred<br>*The preferred (or optimistic) target should reflect what is needed to achieve broader, deeper, quicker global health impact.* | Annotations<br>*For all parameters, include here the source data used and rationale for why this feature is important.* |
|---|---|---|---|
| **Drug interactions** | No significant drug-drug interactions with common antenatal treatments (medicines or supplements) or drugs used in women with preeclampsia (such as anti-hypertensives, antibiotics, magnesium sulfate, tocolytics or corticosteroids), or drugs used for common co-morbidities (including chronic hypertension, type I or II diabetes, obesity, chronic kidney disease or autoimmune disease) | No drug-drug interactions with common antenatal treatments (medicines or supplements) or with drugs used in women with preeclampsia (such as anti-hypertensives, antibiotics, magnesium sulfate, tocolytics or corticosteroids), or drugs used for common co-morbidities (including chronic hypertension, type I or II diabetes, obesity, chronic kidney disease or autoimmune disease). | The treatment must have minimal to no adverse interactions with drugs commonly used in pregnant or postpartum women with preeclampsia |
| **Formulation Dosage & Administration** | Non-invasive (including oral, inhaled, vaginal or transdermal) or parenteral (including intramuscular, intravenous or infusion) Regimen (dose and duration) dependent on clinical response to treatment and severity of preeclampsia. | Oral Regimen (dose and duration) dependent on clinical response to treatment and severity of preeclampsia. | Current interventions for women with preeclampsia are delivered either orally or parenterally, as are experimental treatments being investigated for preeclampsia treatment in ongoing clinical trials [6,28]. Oral administration is preferred, as it would likely be more feasible and acceptable in low-resource settings, particularly in settings with limited capacity to administer and monitor women receiving infusions. |
| **Treatment adherence** | Frequency of discontinuation during therapy <20% | Frequency of discontinuation during therapy <10% | Large multi-center trials of magnesium sulfate and oral antihypertensives during pregnancy have reported discontinuation rates less than 3% [37,38]. Treatment adherence rates do not take into consideration access to healthcare services or supplies. |
| **Stability / Shelf Life** | Stable at 30°C Easy to transport and store. 2-year shelf life in climatic zone IVb (simulated with 30°C and 75% relative humidity). *Biologicals*: cold-chain (2-8°C) acceptable. | Stable at 30°C Easy to transport and store. 3 to 5-year shelf life in climatic zone IVb (simulated with 30°C and 75% relative humidity plus 6-month stability at 40°C and 75% relative humidity). *Biologicals*: cold-chain (2-8°C) acceptable. | Given the greater burden of preeclampsia in LMICs, ease of transport and storage, as well as stability in hotter or humid conditions is a priority [1]. |
| **Product Presentation** | Easy to open and administer. Packaging must aim to protect and preserve the quality of the product and prevent damage to the drugs during transport and storage. *Injectable*: packaging must maintain sterility. | Compact, lightweight, easy to open and administer, sustainable packaging. Packaging must aim to protect and preserve the quality of the product and prevent damage to the drugs during transport and storage. *Injectable*: packaging must maintain sterility. Environmental impact of the packaging should be minimized. | An easy to open and administer presentation will aid in the implementation of the novel treatment, as there will be minimal additional training requirements for healthcare workers. Packaging and design must comply with regulatory guidance from a stringent regulatory authority or WHO standards. |
| **Target Product Registration Pathway(s)** | Approval by at least 1 stringent regulatory authority (e.g., US Food and Drug Administration, European Medicines Agency) Approval from relevant national regulatory authorities will also be required | Approval by at least 1 stringent regulatory authority (e.g., US Food and Drug Administration, European Medicines Agency) Approval from relevant national regulatory authorities will also be required WHO pre-qualification approval obtained | Use of a treatment in a given LMIC will require approval from their national regulatory authority. Product registration pathways are likely to differ for repurposed compared to novel drug treatments. Engaging with regulatory authorities early to discuss potential regulatory pathways and streamline the approval process is advised. |
| **WHO Prequalification** | WHO listed authority application pathways within 12 months of Essential Medicines List (EML) inclusion. | WHO prequalification submission to be made within 12 months of Essential Medicines List (EML) inclusion. | WHO PQ eligibility follows guideline and EML inclusion. |

(*Continued*)

**Table 2.** (Continued)

| | Minimum<br>*The minimal target should be considered as a potential go/no go decision point.* | Preferred<br>*The preferred (or optimistic) target should reflect what is needed to achieve broader, deeper, quicker global health impact.* | Annotations<br>*For all parameters, include here the source data used and rationale for why this feature is important.* |
|---|---|---|---|
| **Primary Target Delivery Channel** | *All*: Antenatal, childbirth and postpartum care settings where where women with preeclampsia are managed and monitored.<br>*Non-invasive*: Staff available to administer oral treatment<br>*Parenteral (including infusion)*: Staff, supplies and equipment available and authorised to administer parenteral treatment | *All*: Antenatal, childbirth and postpartum care settings where women with preeclampsia are managed and monitored.<br>*Oral*: Staff available to administer oral treatment | At a minimum, the treatment (oral or parenteral) would be delivered in settings with the capacity to deliver that treatment and monitor maternal and fetal well-being. |
| **Target Affordable Pricing / Procurement** | Treatment is affordable in the public sector in LMICs | Treatment affordable in the public sector in LMICs<br>Unit cost of treatment is similar to other treatments for women with preeclampsia | Given the burden of preeclampsia in LMICs, affordability of any novel treatments is a high priority and an integral part of access planning.<br>Current treatments for women with preeclampsia (antihypertensive drugs; magnesium sulfate) are generally widely available and affordable. |
| **Expected Financing Source** | Procurement in LMICs financed by national governments, international agencies (including UN organizations), and /or international donors, or private sector | Procurement financed by national governments or private sector | Procurement of medicines for use in pregnancy in LMICs varies between countries, but it may include governments as well as support from international organizations, agencies or funders.<br>For a new treatment, initial support from international organizations or donors may be required.<br>Procurement of effective treatments would ideally be prioritized by national governments. |
| **Volume estimates** | Volumes compatible with incidence of preeclampsia | Same as minimum | The estimated global incidence of preeclampsia is approximately 5%, equating to ~7 million women worldwide each year (though this may be an underestimate) [3]. There are currently no reliable global estimates on the coverage of current preeclampsia treatments in pregnancy, though they are widely used. |

A clear challenge to addressing the global burden of preeclampsia is the need for more "end-to-end" thinking [12]. That is, ensuring that the medicines development pipeline, from pre-clinical research through to Phase III trials and implementation, is aligned with the real-world contexts that clinicians and women face in preventing and managing preeclampsia, particularly the challenges of low-resource settings. In the context of developing TPPs, these challenges were evident in our data related to stability, shelf-life and formulation variables. While novel candidates such as siRNA therapies and monoclonal antibodies are promising, the need for cold-chain transport and storage will likely limit their feasibility in many LMICs. This problem has been well-described in the context of postpartum haemorrhage [43]. IV/IM oxytocin is highly effective when used for postpartum haemorrhage prevention and treatment, however oxytocin requires cold-chain transport and storage. This requirement has not only limited its use in many LMICs, but also contributed to concerningly high rates of low-quality oxytocin in many settings [44]. Consequently, alternative agents such as heat-stable, oral misoprostol with greater side effects for women are necessary to increase coverage of uterotonics. While many novel molecular and antibody therapies require parenteral administration, stakeholders with experience in LMICs raised concerns that even though an injectable medicine

may be effective, repeated injections requiring additional healthcare visits would be a significant barrier for many women, ultimately limiting their potential impact. To strike this balance, both TPPs contain non-invasive or injectable therapies as part of the minimum target for route of administration, and cold chain for biologicals only under the stability/shelf-life domain. It should however be acknowledged that new medicines for preeclampsia prevention and treatment may involve the development of novel products or repurposing of existing medicines for obstetric use. Metformin, a blood glucose lowering drug, shows promise as an effective preventive agent for preeclampsia. A recent clinical trial in women with preterm preeclampsia found metformin significantly prolonged gestation, compared to placebo [45].

Effective medicines to prevent and treat preeclampsia are vital to reducing maternal and newborn mortality and morbidity globally. However, significant challenges remain in place for the development of new maternal medicines. The lack of incentive and willingness to invest in development of new maternal medicines, both by drug developers and by governments, is a key barrier to progress in this field [10,46]. Without strong motivation for stakeholders to coordinate efforts and prioritise maternal and newborn medicines, the development of new medicines for preeclampsia is likely to remain haphazard, impeding more rapid progress [46]. The TPPs we have developed provide a first step in the development of new medicines for preeclampsia that meet specific characteristics defined to meet the global needs of pregnant women and the health care providers caring for them.

These are the first publicly available TPPs designed to address the lack of effective medicines for the prevention and treatment of preeclampsia. They were developed using multiple methods with diverse international participation, in accordance with a pre-specified protocol that was informed by product profile development approaches used in other health fields. Nonetheless, some limitations exist. All interviews were conducted online, and it is possible that face-to-face interviews may have yielded better quality data. While we sought a diversity of participants, it is possible that other members of the maternal health research community or other stakeholder groups may hold different views. Though the exact response rate of the survey cannot be determined (as it was disseminated through multiple channels and listservs), a larger number of responses may have yielded different levels of agreement, however given the already high level of agreement and the strong consensus between the survey results and the in-dept interviews we believe this to be unlikely. Some previous TPP development studies have used two survey rounds to reach consensus with stakeholders on TPP domains [20,23], which can be a useful strategy when there is high disagreement for a large number of domains. In this study, agreement was relatively high and the few areas of disagreement were explained by stakeholder interview data. We consider these TPPs to be "living" documents that may be updated or refined as the preeclampsia R&D field advances, and if further viewpoints or evidence is elucidated.

## Conclusion

There is a lack of new medicines for preeclampsia in the drug development pipeline, and without significant R&D investment the global burden of this condition will likely persist. This TPP development study demonstrated clear agreement amongst diverse stakeholders on the requirements for new medicines in preeclampsia prevention and treatment, with particular emphasis on meeting the real-world needs of low-resource contexts. These TPPs can provide guidance to those involved in maternal medicine research and implementation, including drug developers, clinicians, researchers conducting clinical trials, donors and implementers. They can help stimulate renewed interest, innovation and resources for developing medicines–whether new or repurposed–that can prevent and treat preeclampsia.

## Supporting information

**S1 Appendix. PE TPP interview guide.**
(DOCX)

## Acknowledgments

The authors would like to thank the expert advisory committee to AIM[30] and the stakeholders who participated in this study.

## Author Contributions

**Conceptualization:** Anne Ammerdorffer, A. Metin Gülmezoglu, Joshua P. Vogel.

**Data curation:** Annie Ra Mcdougall, Maya Goldstein, Joshua P. Vogel.

**Formal analysis:** Annie Ra Mcdougall.

**Funding acquisition:** A. Metin Gülmezoglu, Joshua P. Vogel.

**Investigation:** Annie Ra Mcdougall, Andrew Tuttle, Anne Ammerdorffer, A. Metin Gülmezoglu, Joshua P. Vogel.

**Methodology:** Annie Ra Mcdougall, Andrew Tuttle, Maya Goldstein, Anne Ammerdorffer, A. Metin Gülmezoglu, Joshua P. Vogel.

**Project administration:** Anne Ammerdorffer, A. Metin Gülmezoglu.

**Resources:** Anne Ammerdorffer, A. Metin Gülmezoglu.

**Supervision:** A. Metin Gülmezoglu, Joshua P. Vogel.

**Validation:** Annie Ra Mcdougall, Andrew Tuttle.

**Visualization:** Annie Ra Mcdougall, Maya Goldstein.

**Writing – original draft:** Annie Ra Mcdougall.

**Writing – review & editing:** Annie Ra Mcdougall, Andrew Tuttle, Maya Goldstein, Anne Ammerdorffer, A. Metin Gülmezoglu, Joshua P. Vogel.

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
