## [Decision Letter · Decision Letter 0]

14 Sep 2022

PGPH-D-22-01331

Target product profiles for novel medicines to prevent and treat preeclampsia: an expert consensus

Dear Dr. McDougall,

Thank you for submitting your manuscript to PLOS Global Public Health. After careful consideration, we feel that it has merit but does not fully meet PLOS Global Public Health’s publication criteria as it currently stands. Therefore, we invite you to submit a revised version of the manuscript that addresses the points raised during the review process.

Two reviewers have provided constructive feedback and requests for clarification below.

We look forward to receiving your revised manuscript.

Kind regards,

Hannah Tappis, DrPH, MPH

Academic Editor

Journal Requirements:

2. Please change "female” or "male" to "woman” or "man" as appropriate, when used as a noun (see for instance https://apastyle.apa.org/style-grammar-guidelines/bias-free-language/gender).

3.Please provide a/amend your detailed Financial Disclosure statement. This is published with the article. It must therefore be completed in full sentences and contain the exact wording you wish to be published.

   a. Please clarify all sources of funding (financial or material support) for your study. List the grants (with grant       number) or organizations (with url) that supported your study, including funding received from your institution. 

   b. State the initials, alongside each funding source, of each author to receive each grant.

   c. State what role the funders took in the study. If the funders had no role in your study, please state: “The funders       had no role in study design, data collection and analysis, decision to publish, or preparation of the manuscript.”

   d. If any authors received a salary from any of your funders, please state which authors and which funders.

4.lease provide separate figure files in .tif or .eps format.

5. We have noticed that you have uploaded Supporting Information files, but you have not included a list of legends. Please add a full list of legends for your Supporting Information files after the references list. 

Reviewers' comments:

Reviewer's Responses to Questions

**Comments to the Author**

1. Does this manuscript meet PLOS Global Public Health’s publication criteria? Is the manuscript technically sound, and do the data support the conclusions? The manuscript must describe methodologically and ethically rigorous research with conclusions that are appropriately drawn based on the data presented.

Reviewer #1: Yes

Reviewer #2: Yes

2. Has the statistical analysis been performed appropriately and rigorously?

Reviewer #1: N/A

Reviewer #2: Yes

3. Have the authors made all data underlying the findings in their manuscript fully available (please refer to the Data Availability Statement at the start of the manuscript PDF file)?

Reviewer #1: Yes

Reviewer #2: Yes

4. Is the manuscript presented in an intelligible fashion and written in standard English?

Reviewer #1: Yes

Reviewer #2: Yes

5. Review Comments to the Author

Reviewer #1: This manuscript describes the process and related results of an exercise to develop a TPP for medicines for both prevention and treatment of PE/E. Overall, the manuscript is well-written and describes the process and results used in a clear and concise manner. Drug developers and others interested in pharmaceutical innovation may be interested in this work.

Introduction

Page 4, line 117: The authors mention the WHO health product profile directory as the main clearinghouse for TPPs however they neglect other similar efforts being undertaken in other product spaces and by other organizations; see for example Unicef Supply Division; see: https://www.unicef.org/supply/target-product-profiles . It would strengthen their rationale for developing the TPP if they point to other global efforts, not just WHO, even though they may be outside of direct obstetric care.

Methods

Line 154: The use-case scenarios seem to be inconsistent in that the treatment scenario identified who will provide the drug (i.e., skilled health personnel) and the prevention scenario only says that the drug will be administered to pregnant women. Why was there not standardization of the two scenarios?

Line 160-62; citations should be provided for the "templates and guidance produced by several reputable organizations (FDA, Gates Foundation, WHO, PATH and others)"

Line 168: In step 2, 39 stakeholders were invited to participate. Please provide further information about the eligibility criteria for participation, and the sample size (n=39).

Line 195: [Add dates here] needs to be corrected

Line 207: Please explain Qualtrics. "The survey was conducted using Qualtrics and...".

Line 218: Survey invitations were sent to approximately 270 individuals using several databases however the databases used were highly skewed towards previous participation in some kind of WHO activity. I have been working explicitly in the PE/E field for several years and never heard about this "international survey" so it makes me wonder how far-reaching the invitation really was. For example, was the invitation disseminated via the RHSC Maternal Health Caucus and similar implementing agency channels?

Line 232: Invitation for public consultation was disseminated via the Burnet Institute Twitter account. How many responses were captured via this "public consultation"? This seems to be a rather nominal way to obtain public input. For this reason, it would be helpful to understand the extent of "public" comment. Why was this method selected as a way to gain public comment?

Results

Throughout this section, it would be helpful to put sample sizes where results are being reported. Also, it would be helpful to keep the order the same as when it was described in the methods section. In the current manuscript, it is difficult to track the various data collection efforts so some clarity is required.

Line 289: Figure 2 should be labeled better; what is A, B, C D? The narrative that describes Figure 2 is confusing.

LIne 296: I think it would be easier to follow this discussion if the final TPPs for both prevention and treatment (Tables 1 and 2) were presented first since the narrative related directly to the TPPs.

Discussion

LIne 409: PE/E diagnostics have already benefitted from TPPs for specific product development efforts; see for example: https://media.path.org/documents/PrCr_11.20_.pdf

line 480: I am not sure many people know what "triallists" are? Perhaps rephrase?

Reviewer #2: This is a well-conducted study with the objective to develop two new TPPs for medicines to prevent and treat preeclampsia. The manuscript is well-written and adds important information in this area.  

6. PLOS authors have the option to publish the peer review history of their article (what does this mean?). If published, this will include your full peer review and any attached files.

**Do you want your identity to be public for this peer review?** For information about this choice, including consent withdrawal, please see our Privacy Policy.

Reviewer #1: No

Reviewer #2: No

---

## [Editor Report · Decision Letter 1]

25 Oct 2022

Target product profiles for novel medicines to prevent and treat preeclampsia: an expert consensus

PGPH-D-22-01331R1

Dear Dr McDougall,

We are pleased to inform you that your manuscript 'Target product profiles for novel medicines to prevent and treat preeclampsia: an expert consensus' has been provisionally accepted for publication in PLOS Global Public Health.

Best regards,

Hannah Tappis, DrPH, MPH

Academic Editor